# DRAG: Data Reconstruction Attack using Guided Diffusion

## Abstract

With the rise of large foundation models, split inference (SI) has emerged as a popular computational paradigm for deploying models across lightweight edge devices and cloud servers, addressing both data privacy and computational cost concerns. However, most existing data reconstruction attacks have focused on smaller classification models like ResNet, leaving the privacy risks of foundation models in SI settings largely unexplored. To address this gap, we propose a novel data reconstruction attack based on guided diffusion, which leverages the rich prior knowledge embedded in a latent diffusion model (LDM) pretrained on a large-scale dataset. Our method performs iterative reconstruction on the LDM's learned image manifold, effectively generating high-fidelity images closely resembling the original data from their intermediate representations (IR). Extensive experiments demonstrate that our approach significantly outperforms state-of-the-art methods, both qualitatively and quantitatively, in reconstructing data from deep-layer IRs of the vision foundation model. The results highlight the urgent need for more robust privacy protection mechanisms for large models in SI scenarios.

## 1 Introduction

The rapid development of deep learning has revolutionized various aspects of daily life—from AI assistants to autonomous vehicles. However, the substantial computational resources required by these emerging models often hinder their deployment on edge devices. Therefore, offloading intensive computation to cloud servers become a popular alternative. Following this paradigm, Split inference (SI) (Kang et al., 2017) has emerged as one of the most promising solutions, as it balances both computational and privacy concerns. This approach enables efficient utilization of cloud resources, reduces the computational burden on local devices, and facilitates the integration of complex models into everyday technologies by partitioning neural network computations between edge devices and cloud servers, with data processed locally before being sent to the server.

Despite its advantages, recent studies (He et al., 2019; Dong et al., 2021; Li et al., 2024; Xu et al., 2024; Sa et al., 2024) have uncovered significant privacy risks associated with SI, particularly in the form of data reconstruction attacks (DRA). In DRA, adversaries attempt to reconstruct clients' input data by exploiting the exchanged intermediate representations (IR) between clients and servers, posing serious threats that break user's privacy.

However, the growing adoption of more powerful models, such as Vision Transformers (Dosovitskiy et al., 2020), raises concerns about the effectiveness of existing defenses. Vision Transformers have demonstrated superior performance across various vision tasks and are widely used in modern applications. Despite this, the privacy implications of deploying these models in SI settings remain underexplored.

In this paper, we address this gap by investigating privacy leaks in vision transformers in the context of SI. We propose a novel attack based on guided diffusion that effectively utilizes the prior knowledge captured by large latent diffusion models (LDM) Rombach et al. (2022) pretrained on large-scale datasets (e.g., Stable Diffusion) to reconstruct input data from deep-layer IR. Leveraging this prior knowledge, we successfully invert IR back to the original input data across various natural image datasets, revealing a critical privacy vulnerability in the SI framework. Additionally, we evaluate our attack on models equipped with existing defenses (Singh et al., 2021; Vepakomma et al.,

2020) and show that input data can still be successfully reconstructed from deep-layer IR despite the defenses. Our key contributions are summarized as follows:

- We propose a novel attack that exploits the prior knowledge captured by LDMs to reconstruct input data from deep-layer IR.
- We show our attack can easily reconstruct the data from the widely used vision foundation models, specifically CLIP (Radford et al., 2021), demonstrating that the privacy threat exists even in general-purpose, widely used vision encoders.
- We explore different defense strategies tailored for vision transformers to mitigate the threat of privacy leakage.

## 2 RELATED WORK

### 2.1 SPLIT INFERENCE / COLLABORATIVE INFERENCE

Split inference (SI) (Kang et al., 2017) is a method aimed at speeding up inference and/or reducing power consumption on end-point devices while ensuring data privacy. Unlike the approach of performing inference completely on the cloud, which requires sending raw data to the server, SI enhances privacy by sending only transformed, non-trivially interpretable IR to the cloud. In SI, the model $f$ is split into two part: the client model $f_c : \mathcal{X} \to \mathcal{H}$, deployed on the edge device, and the server model $f_s : \mathcal{H} \to \mathcal{Y}$, deployed on the cloud. During inference, the private data $\mathbf{x}_{\text{private}}$ is first processed by $f_c$ on the edge, producing the "smashed" data tensor $\mathbf{H}_{\text{private}} = f_c(\mathbf{x}_{\text{private}})$. This IR is then sent to the cloud, where $f_s$ completes the remaining computation to obtain the result $y_{\text{private}} = f_s(\mathbf{H}_{\text{private}})$. SI therefore addresses the constraints of limited computational resources on edge-side while preventing direct exposure of private data to the cloud.

### 2.2 DATA RECONSTRUCTION ATTACK

One line of prior research focuses on the privacy of users' input data. In the context of SI, an adversary may carry out data reconstruction attacks to extract private information by reconstructing the input data. According to He et al. (2019), data reconstruction attacks can be classified into three types: 1) white-box attacks, 2) black-box attacks, and 3) query-free attacks.

For the white-box attacks, He et al. (2019) first introduced *regularized Maximum Likelihood Estimation* (rMLE), which optimizes the zero-initialized input $\mathbf{x}$ to minimize the distance between the IR of the reconstructed input $\mathbf{H} = f_c(\mathbf{x})$ and $\mathbf{H}_{\text{private}}$ obtained during the message exchanges. Total Variation (Rudin et al., 1992) loss is applied as an image prior to ensure the optimization process not only minimizes the distance but also produces results that appear natural to human perspective. Singh et al. (2021) further improved reconstruction quality by incorporating a deep image prior (Ulyanov et al., 2018), resulting in the *Likelihood Estimation* (LM) approach. Dong et al. (2021), on the other hand, considers to improve the training of inverse network (He et al., 2019) by applying a cycle loss. Li et al. (2024) proposed *GAN-based Latent Space Search* (GLASS), which constrains the search space of $\mathbf{x}$ by optimizing the latent code in StyleGAN2 (Karras et al., 2020), achieving high quality reconstructions and circumventing several defenses (He et al., 2019; Singh et al., 2021; Titcombe et al., 2021; Mireshghallah et al., 2020; Li et al., 2021; Osia et al., 2020).

However, these works primarily focus on evaluating small CNN models, such as ResNet18 (He et al., 2016), leaving their effectiveness against more advanced architecture, such as Vision Transformer (ViT) (Dosovitskiy et al., 2020), unclear.

### 2.3 DIFFUSION MODELS

In recent years, diffusion models (Ho et al., 2020) have demonstrated remarkable capabilities in generating realistic images. Several methods (Dhariwal & Nichol, 2021; Ho & Salimans, 2022) have been developed to enable controlled content generation. The LDM (Rombach et al., 2022) further expanded the ability to generate high-resolution, diverse images, with subsequent works (Ramesh et al., 2022; Zhang et al., 2023) extending control within the latent diffusion framework. Beyond conditional generation through training, another line of research (Chung et al., 2023; Bansal

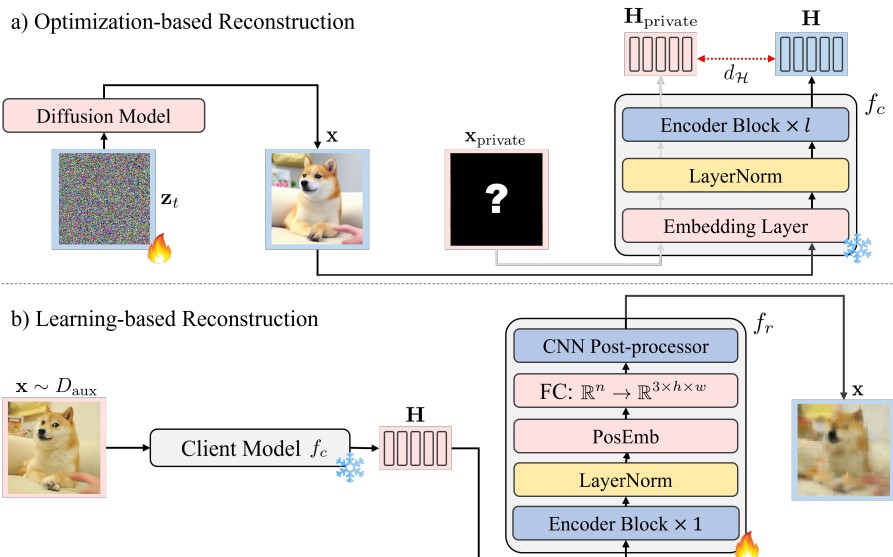

Figure 1: An illustration of our reconstruction pipeline. The server acts as an honest-but-curious participant, attempting to reconstruct the input using either optimization-based or learning-based reconstruction. We analyze the threat using CLIP-ViT-B/16 with split layers at $l = \{0, 3, 6, 9, 12\}$.

et al., 2024; He et al., 2024) investigates leveraging the prior knowledge of diffusion models without additional model training.

## 3 METHODOLOGY

### 3.1 THREAT MODEL

Following prior works (He et al., 2019; Singh et al., 2021; Dong et al., 2021; Li et al., 2024; Sa et al., 2024), we examine the privacy risks in the context of split inference (Kang et al., 2017). In this scenario, a client with limited computation resources seeks to offload the inference computation to a cloud server without exposing the private data $\mathbf{x}_{\text{private}}$. To achieve this, the client decomposes the model $f = f_s \circ f_c$ into two parts, only deploying $f_s$ to the server. During inference, the client first feeds the data $\mathbf{x}_{\text{private}}$ to obtain the hidden state $\mathbf{H}_{\text{private}} = f_c(\mathbf{x}_{\text{private}})$, then sends $\mathbf{H}_{\text{private}}$ to the server. The server finishes the remaining computation tasks and sends output $y = f_s(\mathbf{H}_{\text{private}})$ to the client.

The server, acting as an honest-but-curious participant, aims to reconstruct the private data $\mathbf{x}_{\text{private}}$ from the hidden state $\mathbf{H}_{\text{private}}$. We then assess the privacy threat posed by adversaries with white-box access to the model. In this scenario, the adversary is assumed to have full knowledge of the architecture and parameters of the client's model $f_c$. Specifically, the client uses a foundation model as the backbone for their applications, keeping its parameters frozen for downstream tasks. Following the taxonomy of (He et al., 2019), there are two ways to formulate the problem: optimization based and learning based.

**Optimization Based.** The adversary aims to find input data $\mathbf{x}$ whose hidden state $\mathbf{H}$ closely matches $\mathbf{H}_{\text{private}}$ through optimization:

$$\min_{\mathbf{x}} d_{\mathcal{H}}(f_c(\mathbf{x}), f_c(\mathbf{x}_{\text{private}})) + \lambda R_{\mathcal{I}}(\mathbf{x}), \tag{1}$$

where $d_{\mathcal{H}}$ measures the distance between the IR of reconstruction $\mathbf{x}$, $\mathcal{I}$ is the natural image manifold which $\mathbf{x}$ lies on, $R_{\mathcal{I}}$ represents the regularization terms that ensuring that $\mathbf{x}$ appears natural to human perception, and $\lambda \geq 0$ controls the weight of regularization. Ultimately, the adversary's goal is to obtain a reconstruction $\mathbf{x} \approx \mathbf{x}_{\text{private}}$.

**Learning Based.** The adversary may attempt to infer $\mathbf{x}_{\text{private}}$ by training a model $f_r$ to decode $\mathbf{H}_{\text{private}}$ back to data space $\mathcal{X}$ using an auxiliary dataset $D_{\text{aux}}$.

$$\min_{f_r} \mathbb{E}_{\mathbf{x} \sim D_{\text{public}}} ||f_r(f_c(\mathbf{x})) - \mathbf{x}||_2 \tag{2}$$

## 3.2 GUIDED DIFFUSION DATA RECONSTRUCTION ATTACK

According to Song et al. (2021), the noise prediction network $\epsilon_\theta$ in the diffusion model can be interpreted as a score function, i.e. $\epsilon_\theta(\mathbf{x}_t) \approx \nabla_{\mathbf{x}} \log p_t(\mathbf{x})$. Building on this idea, we propose leveraging the trained noise predictor as an image prior. Specifically, the image is reconstructed by a generative model based on a conditional reverse-time SDE:

$$d\mathbf{x} = \{\mathbf{f}(\mathbf{x}, t) - g(t)^2 [\nabla_{\mathbf{x}} \log p_t(\mathbf{x}) + \nabla_{\mathbf{x}} \log p_t(\mathbf{y}|\mathbf{x})]\}dt + g(t)\bar{\mathbf{w}}. \tag{3}$$

In the case of class-conditioned generation, $p_t(\mathbf{y}|\mathbf{x})$ can be approximated by a classifier model training with data pairs with noisy input $(\mathbf{x}_t, \mathbf{y})$. However, according to Chung et al. (2023), for noisy inverse problem (solve $\mathbf{x}$ given $\mathbf{y} = \mathcal{A}(\mathbf{x}) + n$, where $\mathcal{A}$ is the measurement operator, $\mathbf{y}$ is the measurement to $\mathbf{x}$, and $n$ is Gaussian noise), $p_t(\mathbf{y}|\mathbf{x})$ is intractable because $\mathcal{A}$ is only defined for $t = 0$. To exploit $p_0(\mathbf{y}|\mathbf{x})$, they propose the use of Tweedie estimation to predict $\mathbf{x}_0$ from $t > 0$, denoted as $\hat{\mathbf{x}}_0$:

$$\hat{\mathbf{x}}_0 = \frac{\mathbf{x}_t - \sqrt{1 - \alpha_t} \cdot \epsilon_\theta(\mathbf{x}_t, t)}{\sqrt{\alpha_t}} \tag{4}$$

Furthermore, Bansal et al. (2024) propose *Universal Guidance Diffusion* (UGD), which leverages gradient from various models trained exclusively on clean images as a conditioning signal in the sampling process. Specifically, UGD incorporates a guidance term into the predicted noise during the denoising process, extending the concept of classifier guidance (Ho & Salimans, 2022):

$$\hat{\epsilon}_\theta(\mathbf{x}_t) = \epsilon_\theta(\mathbf{x}_t) + w \cdot \sqrt{1 - \alpha_t} \cdot \nabla_{\mathbf{x}_t} \ell(\mathbf{x}_t), \tag{5}$$

where $w$ is the guidance scale. These methods extend conditional generation using various deep learning models trained solely on clean images. Building on this concept, we define the loss $\ell(\mathbf{x}_t)$ to approximate $\log p_t(\mathbf{y}|\mathbf{x})$, where the loss consists of the distance in the hidden state $d_{\mathcal{H}}$ and an auxiliary regularization term $R$.

$$\ell(\mathbf{x}_t) = d_{\mathcal{H}}(f_c(\hat{\mathbf{x}}_0), \mathbf{H}_{\text{private}}) + \lambda R(\hat{\mathbf{x}}_0). \tag{6}$$

While using the LDM, we replace noisy sample $\mathbf{x}_t$ as $\mathbf{z}_t$, and the data $\mathbf{x}_t = \mathcal{D}(\mathbf{z}_t)$ can be obtained by decoding it with the corresponding latent decoder.

---

**Algorithm 1** DRAG: Data Reconstruction Attack using Guided diffusion

  **Parameter:** Recurrent steps $k$, guidance strength $w$, diffusion step $T$
  **Required:** LDM ($\epsilon_\theta, \mathcal{E}, \mathcal{D}, \{\alpha_t\}_{t=1}^T$), to-be-inverted hidden state $\mathbf{H}$
  **if** hidden state decoder $f_r$ is available **then**
    $\mathbf{z}_T \leftarrow \text{DDPM}(\mathcal{E}(f_r(\mathbf{H}_{\text{private}})), T)$              ▷ Init. from decoder's reconstruction
  **else**
    $\mathbf{z}_T \sim \mathcal{N}(\mathbf{0}, \mathbf{I})$
  **end if**
  **for** $t = T, T-1, \ldots, 1$ **do**
    **for** $n = 1, 2, \ldots, k$ **do**
      $\hat{\mathbf{z}}_0 \leftarrow (\mathbf{z}_t - \sqrt{1 - \alpha_t} \cdot \epsilon_\theta(\mathbf{z}_t))/\sqrt{\alpha_t}$           ▷ Tweedie's estimation
      $\hat{\mathbf{x}}_0 \leftarrow \mathcal{D}(\hat{\mathbf{z}}_0)$               ▷ Decode the latent to image
      $g \leftarrow \nabla_{\mathbf{z}_t}(d_{\mathcal{H}}(f_c(\hat{\mathbf{x}}_0), \mathbf{H}) + R(\hat{\mathbf{x}}_0))$
      $\hat{\epsilon}_\theta(\mathbf{z}_t) \leftarrow \epsilon_\theta(\mathbf{z}_t) + w \cdot \sqrt{1 - \alpha_t} \cdot g$      ▷ Guided diffusion (Eq. (5))
      $\mathbf{z}_{t-1} \leftarrow \text{DDIM}(\mathbf{z}_t, \hat{\epsilon}_\theta, t)$           ▷ DDIM sampling
      $\epsilon' \sim \mathcal{N}(\mathbf{0}, \mathbf{I})$                ▷ DDPM diffusion
      $\mathbf{z}_t \leftarrow \sqrt{\alpha_t/\alpha_{t-1}} \cdot \mathbf{z}_{t-1} + \sqrt{1 - \alpha_t/\alpha_{t-1}} \cdot \epsilon'$
    **end for**
  **end for**
  **return** $\mathbf{x} = \mathcal{D}(\mathbf{z}_0)$

---

### 3.3 LEARNING-BASED DATA RECONSTRUCTION ATTACK

When the split point is too deep, optimizing latent $\mathbf{z}_t$ from randomly initialized noise cannot guarantee high-quality reconstructions. To address this, we propose using an auxiliary dataset collected from publicly available resources to train a lightweight model $f_r : \mathcal{H} \rightarrow \mathcal{X}$ by minimizing Eq. (2). This model consists of a transformer encoder block, a linear layer, a learnable position embedding layer, and a CNN-based post-processing module. This setup resembles the decoder architecture in ViT-MAE (He et al., 2022). Details can be found in Fig. 1 (b).

The coarse reconstruction result from $f_r$ serves as a better initialization for the latent $\mathbf{z}_t$ during optimization-based data reconstruction. Specifically, we apply DDPM diffusion, adding random noise with a small timestep $t \leq T$, to project the coarse reconstruction onto an editable manifold. The complete pseudocode is provided in Algorithm 1.

## 4 EXPERIMENT SETTING

### 4.1 DATASETS

We evaluate our method using three datasets: (1) FFHQ (Karras et al., 2019), (2) MSCOCO (Lin et al., 2014), and (3) ImageNet-1K (Deng et al., 2009). All images are resized and center-cropped to $224 \times 224$ to match the input size of the target model. For each dataset, the training split is divided into two distinct, equal-sized, non-overlapping parts: a private portion $D_{\text{private}}$ and a public portion $D_{\text{public}}$. The target model is fine-tuned exclusively on $D_{\text{private}}$, while the attacker model is trained solely on $D_{\text{public}}$. The model's utility and privacy index are evaluated using the validation set $D_{\text{val}}$, which is not used in fine-tuning or attacker model training. For the optimization-based attack, we randomly sample 10 images from the validation split of each dataset as the target images. For the learning-based attack, we use the entire validation split as the evaluation target.

### 4.2 TARGET MODEL

We aim to reconstruct data from the widely-used vision encoder CLIP-ViT-B/16[1] (Radford et al., 2021), which has demonstrated strong adaptability and zero-shot capabilities across various vision tasks (Rao et al., 2022; Mokady et al., 2021). We evaluate the attack under three conditions, 1) the model is frozen from the pretrained checkpoint, or fine-tuned by state-of-the-art defensive algorithm 2) NoPeek (Vepakomma et al., 2020) or 3) DISCO (Singh et al., 2021). These two defense algorithms, highlighted in GLASS(Li et al., 2024), have shown superior privacy-preserving performance compared to other defense methods. To quantitatively assess model utility after applying these defenses, we select ImageNet-1K image classification as the primary task. To adapt the model for downstream tasks, we first perform linear probing on the ImageNet-1K data, followed by fine-tuning the entire model using the selected defensive algorithms.

### 4.3 BASELINE AND METRICS

We compare our method with rMLE (He et al., 2019), LM (Singh et al., 2021) and GLASS (Li et al., 2024). We choose to evaluate the performance on MSE, PSNR (Horé & Ziou, 2010), SSIM (Wang et al., 2004), LPIPS (Zhang et al., 2018), and image similarity using DINO ViT-S/16[2] (Caron et al., 2021) as the similarity evaluation model.

### 4.4 ATTACKER MODELS, DISTANCE FUNCTION AND REGULARIZATION

We use the official release of Stable Diffusion v1.5[3] as the image prior, which was previously available on Hugging Face but became closed-source as of August 2024. During the reconstruction, we use DDIM (Song et al., 2020) as the sampling strategy to gradually denoising $\mathbf{z}_T$ to $\mathbf{z}_0$. Unlike previous works (He et al., 2019; Singh et al., 2021; Li et al., 2024), which focus on reconstructing

---

[1] https://huggingface.co/openai/clip-vit-base-patch16

[2] https://huggingface.co/facebook/dino-vits16

[3] A mirror of the checkpoint: https://huggingface.co/stable-diffusion-v1-5/stable-diffusion-v1-5

Figure 2: Reconstruction results for the target model CLIP-ViT-B/16.

data from CNN models, we use the average token-wise cosine distance as the distance metric $d_{\mathcal{H}}$:

$$d_{\mathcal{H}}(\mathbf{H}_1, \mathbf{H}_2) = \frac{1}{N} \sum_{i=1}^{N} 1 - \frac{\langle \mathbf{H}_1[i,:], \mathbf{H}_2[i,:] \rangle}{||\mathbf{H}_1[i,:]|| \cdot ||\mathbf{H}_2[i,:]||}, \tag{7}$$

where $N$ represents the number of tokens. To prevent the latent $\mathbf{z}_t$ from being updated outside the distribution of noise predictor $\epsilon_\theta$, we apply regularization to the reconstruction result:

$$R(\mathbf{x}) = \frac{1}{\text{CHW}} (\lambda_1 \mathbf{x}^2 + \lambda_2 \max(\mathbf{x}^2 - 1, 0)), \tag{8}$$

where the $\ell_2$ regularization is also applied in Yin et al. (2020).

## 5 RECONSTRUCTION PERFORMANCE

### 5.1 RECONSTRUCTING FROM FROZEN FOUNDATION MODEL

We present the results in Fig. 2 and Table 1. Our method performs comparable to GLASS, which also employs a data-driven image prior. However, due to the domain limitations of the pretrained GAN, GLASS is unable to reconstruct data from MSCOCO and ImageNet. In contrast, our method leverages the broader coverage and capacity of modern diffusion models, enabling successful reconstruction across a wider range of domains. Additionally, we compare these methods using the FFHQ dataset, where both the target model and attacker operate within the same domain, aligning with GLASS's assumptions. The results are shown in Table 5.

Our method offers notable advantages in reconstructing data from deep layers. While rMLE and LM perform well on shallow layers, their performance declines significantly after Layer 9 and Layer 12, respectively. In contrast, our method maintains strong performance in deep layers, achieving better SSIM, LPIPS, and image similarity metrics. Furthermore, rMLE and LM struggle to reconstruct images with minimal information, such as those compressed to a single token, as seen in the split point "Embedding" in Fig. 2. This shows the advantage of using image prior learned by diffusion model over human-defined priors, especially in challenging reconstruction scenarios. Although our method scores lower on shallow layers, the reconstructed images—while lacking some details (e.g., the floor and background in the first reconstruction target from the MSCOCO dataset)—still preserve a similar high-level structure, demonstrating successful data reconstruction.

### 5.2 ATTACKING DEFENSIVE FINE-TUNED MODEL

We also evaluate the robustness of our method on models equipped with privacy-preserving defenses. As shown in Table 2 and Fig. 3a) - Fig. 3c), our method achieves superior SSIM, LPIPS, and image similarity on three model checkpoints protected by DISCO. However, when the model is fine-tuned with NoPeek, only our methods incorporating an auxiliary decoder perform well. We suspect this is due to NoPeek's use of gradient obfuscation, which interferes with the guided diffusion sampling process.

Table 1: Performance of optimization-based attacks on CLIP-ViT-B/16 at different split points, with no defenses applied.

| Split Point | Method | MSE ($\downarrow$) | PSNR ($\uparrow$) | SSIM ($\uparrow$) | LPIPS ($\downarrow$) | DINO ($\uparrow$) |
|---|---|---|---|---|---|---|
| Layer 0 | rMLE | 0.0073 | 22.6780 | 0.7964 | 0.0709 | 0.9712 |
| | LM | **0.0022** | **28.6115** | **0.9221** | **0.0237** | **0.9903** |
| | GLASS | 0.0094 | 23.2332 | 0.7019 | 0.1676 | 0.7282 |
| | DRAG | 0.0061 | 23.1062 | 0.7178 | 0.1708 | 0.7310 |
| | DRAG++ | 0.0060 | 23.1654 | 0.7296 | 0.1643 | 0.7386 |
| Layer 3 | rMLE | 0.0114 | 20.4385 | 0.7817 | 0.0913 | 0.9705 |
| | LM | **0.0018** | **29.2470** | **0.9081** | **0.0206** | **0.9923** |
| | GLASS | 0.0241 | 19.4969 | 0.5965 | 0.1594 | 0.8318 |
| | DRAG | 0.0106 | 20.6065 | 0.6407 | 0.1389 | 0.8581 |
| | DRAG++ | 0.0097 | 20.9782 | 0.6556 | 0.1299 | 0.8703 |
| Layer 6 | rMLE | 0.0242 | 16.5919 | 0.5459 | 0.2608 | 0.8875 |
| | LM | **0.0106** | **21.7779** | **0.7222** | **0.0784** | **0.9733** |
| | GLASS | 0.0536 | 14.9205 | 0.4179 | 0.2890 | 0.7648 |
| | DRAG | 0.0216 | 17.4993 | 0.5081 | 0.1812 | 0.8680 |
| | DRAG++ | 0.0167 | 18.6246 | 0.5503 | 0.1819 | 0.8803 |
| Layer 9 | rMLE | 0.0389 | 14.3838 | 0.3434 | 0.5130 | 0.7159 |
| | LM | 0.0302 | 16.9228 | **0.5351** | **0.2137** | **0.9063** |
| | GLASS | 0.1130 | 11.3410 | 0.3101 | 0.4163 | 0.6590 |
| | DRAG | 0.0366 | 15.3397 | 0.4449 | 0.2467 | 0.8363 |
| | DRAG++ | **0.0212** | **17.4733** | 0.4939 | 0.2943 | 0.8148 |
| Layer 12 | rMLE | 0.0519 | 13.1381 | 0.2831 | 0.5900 | 0.6524 |
| | LM | 0.0998 | 10.4085 | 0.3303 | 0.6024 | 0.4248 |
| | GLASS | 0.1828 | 8.1962 | 0.2210 | 0.5699 | 0.4362 |
| | DRAG | 0.0683 | 12.7229 | 0.3731 | 0.3838 | 0.6899 |
| | DRAG++ | **0.0280** | **16.0945** | **0.4450** | **0.4316** | **0.7228** |
| Embedding | GLASS | 0.1087 | 10.0151 | 0.2003 | **0.5494** | 0.4918 |
| | DRAG | **0.0967** | **10.5387** | **0.2507** | 0.5849 | **0.6993** |

## 5.3 SHUFFLE (AND DROP TOKEN) DEFENSE

ViTs naturally exhibit an adaptive computation capability compared to CNNs, allowing them to reduce inference time by identifying and halting the forward propagating of redundant tokens. Previous work (Yin et al., 2022) explores strategies to reduce redundant tokens in intermediate layers. From a privacy protection perspective, shuffling patch tokens complicates data reconstruction for attackers, as the loss function for guiding reconstruction (Eq. (7)) is sensitive to token order. Clients can even retain all patch tokens without compromising model performance. For tasks where token order is irrelevant (e.g., classification), shuffling patch tokens provides a straightforward defense against data reconstruction attacks. Moreover, this method is simple for clients to implement, requiring only token shuffling before transmission to the server, with minimal memory copying overhead.

We further evaluate the attack performance against the shuffling defense in a white-box setting. To simulate the scenario of token dropping, we design the following protocol: the client shuffles patch tokens and randomly drops $rN$ patch tokens before sending them to the server, where $r$ is the ratio of the tokens being dropped (see Fig. 4). Note that a realistic implementation would be more complicated than this evaluation protocol, as clients may combine multiple strategies to both reduce inference time and enhance data privacy.

As noted in Darcet et al. (2024), tokens retain information about their original positions, which can be inferred using a linear layer. Based on this observation, we train a 2-layer MLP classifier to predict the probability that a token $\mathbf{H}_{i,:}$ is originally at position $\arg\max p_\theta(\mathbf{H}_{i,:})$. Once trained, the

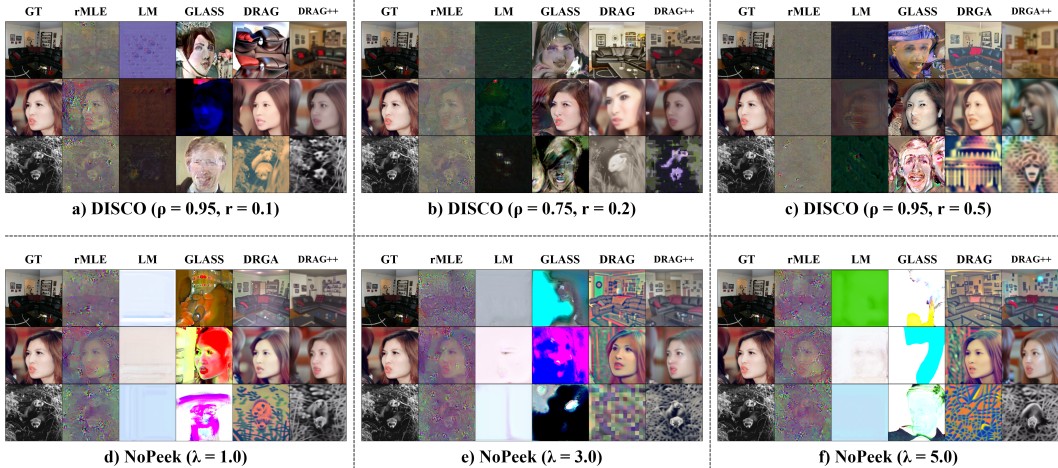

Figure 3: Reconstruction data against the models deployed with defenses.

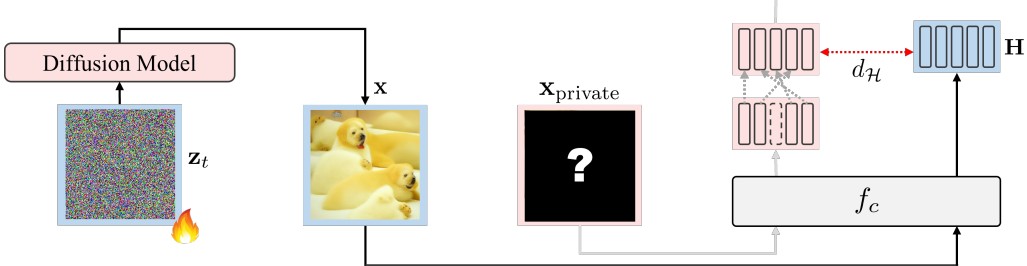

Figure 4: Obfuscating by randomly permuting tokens. While the attacker is unaware that the patch tokens are permuted, the loss gives an incorrect guidance to reconstruct the data.

classifier allows us to reorder the patch tokens by solving a bipartite matching problem, maximizing the joint probability using the Hungarian algorithm (Kuhn, 1955).

In Fig. 5, we present the reconstruction results under three configurations: 1) tokens are randomly permuted, and the adversary is unaware of the permutation, 2) the adversary employs a token position classifier to reorder the tokens, and 3) the client drops 50% of the patch tokens before sending them to the server, leaving the adversary to infer their correct placement. The experiments are conducted with model splitting at layer 12. The position prediction model achieves 12.78% top-1 accuracy in predicting token positions, with an average $\ell_1$ distance of 3.077 from the correct position, as evaluated on ImageNet-1K. Comparing Fig. 5a) and Fig. 5b), we observe that rMLE and LM fail to reconstruct the target images, whereas the normal configuration succeeds. For GLASS and DRAG, the reconstruction performance is weaker, but some reconstructed results retain key properties of the original images.

## 6    CONCLUSION

This work reveals significant privacy risks in Split Inference (SI) with large vision foundation models like Vision Transformers, extending beyond previous attacks on smaller models like ResNet18. We propose a novel data reconstruction attack leveraging LDMs pretrained on large-scale datasets. Our method generates high-fidelity images from IR and outperforms state-of-the-art approaches in reconstructing data from deep-layer IR. These findings underscore the need for stronger defenses to protect privacy when deploying transformer-based models in SI settings.

Table 2: Performance of the optimization-based attack against various defenses. The model is split at the output of **layer 12**.

| Defense | Method | MSE ($\downarrow$) | PSNR ($\uparrow$) | SSIM ($\uparrow$) | LPIPS ($\downarrow$) | DINO ($\uparrow$) |
|---------|--------|------|------|------|-------|------|
| a) DISCO | rMLE | 0.0564 | 12.7443 | 0.2831 | 0.5973 | 0.6067 |
| ($\rho = 0.95$, $r = 0.1$) | LM | 0.1193 | 10.4978 | 0.3101 | 0.6340 | 0.3009 |
| (Acc = 79.61%) | GLASS | 0.1372 | 9.2676 | 0.2454 | 0.5444 | 0.4398 |
| | DRAG | 0.0755 | 12.4247 | 0.3670 | **0.4113** | 0.6710 |
| | DRAG++ | **0.0474** | **14.3449** | **0.4099** | 0.4603 | **0.6973** |
| b) DISCO | rMLE | 0.0629 | 12.3146 | 0.2687 | 0.6587 | 0.4912 |
| ($\rho = 0.75$, $r = 0.2$) | LM | 0.1143 | 9.7468 | 0.2783 | 0.6684 | 0.2329 |
| (Acc = -%) | GLASS | 0.1372 | 9.1868 | 0.1981 | 0.5641 | 0.4168 |
| | DRAG | 0.0927 | 11.3654 | 0.3124 | **0.4914** | 0.5690 |
| | DRAG++ | **0.0556** | **13.2949** | **0.3767** | 0.5191 | **0.6347** |
| c) DISCO | rMLE | **0.0720** | 11.7121 | 0.2517 | 0.7957 | 0.0079 |
| ($\rho = 0.95$, $r = 0.5$) | LM | 0.0967 | 10.6190 | 0.3134 | 0.6456 | 0.0627 |
| (Acc = -%) | GLASS | 0.1345 | 9.3663 | 0.1992 | 0.5673 | 0.1854 |
| | DRAG | 0.0779 | **12.0319** | **0.3343** | **0.4761** | **0.4366** |
| | DRAG++ | 0.0973 | 10.5521 | 0.2934 | 0.6751 | 0.1821 |
| d) NoPeek | rMLE | 0.0673 | 12.1172 | 0.2488 | 0.6604 | 0.4729 |
| ($\lambda = 1.0$) | LM | 0.2863 | 5.9663 | 0.3011 | 0.7159 | 0.2272 |
| (Acc = 79.28%) | GLASS | 0.2049 | 7.4435 | 0.1730 | 0.6503 | 0.3866 |
| | DRAG | 0.1009 | 10.6299 | 0.2668 | 0.5563 | 0.5300 |
| | DRAG++ | **0.0351** | **15.0698** | **0.4112** | **0.4750** | **0.6799** |
| e) NoPeek | rMLE | 0.0725 | 11.6398 | 0.2096 | 0.6938 | 0.3863 |
| ($\lambda = 3.0$) | LM | 0.2677 | 6.5185 | 0.2967 | 0.7377 | 0.2113 |
| (Acc = 78.67%) | GLASS | 0.2076 | 7.3598 | 0.1678 | 0.6751 | 0.3367 |
| | DRAG | 0.0934 | 10.6228 | 0.2418 | 0.5885 | 0.4840 |
| | DRAG++ | **0.0446** | **13.8540** | **0.3811** | **0.4708** | **0.6823** |
| f) NoPeek | rMLE | 0.0741 | 11.5433 | 0.2059 | 0.7048 | 0.3747 |
| ($\lambda = 5.0$) | LM | 0.2387 | 6.6693 | 0.2840 | 0.7917 | 0.2099 |
| (Acc = 77.88%) | GLASS | 0.2431 | 6.7415 | 0.1685 | 0.6615 | 0.3492 |
| | DRAG | 0.1001 | 10.5774 | 0.2511 | 0.5875 | 0.4399 |
| | DRAG++ | **0.0474** | **13.6302** | **0.3630** | **0.4721** | **0.6779** |

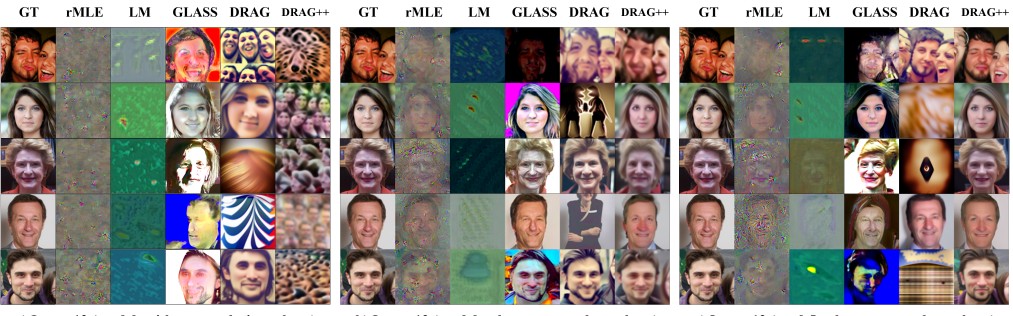

a) Layer 12 (r = 0.0, without reordering tokens)    b) Layer 12 (r = 0.0, adversary reorders tokens)    c) Layer 12 (r = 0.5, adversary reorders tokens)

Figure 5: Illustration of the reconstruction result against random shuffle defense.

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

## A    PRIOR RECONSTRUCTION ALGORITHM

**rMLE.** He et al. (2019) first proposed an optimization-based reconstruction approach that reconstructs $\mathbf{x}_{\text{private}}$ by iteratively updating $\mathbf{x}$ to minimize the distance between $\mathbf{H}_{\text{private}}$ and $\mathbf{H}$, incorporating Total Variation (Rudin et al., 1992) as an image prior:

$$\min_{\mathbf{x}} d_{\mathcal{H}}(f_c(\mathbf{x}), \mathbf{H}_{\text{private}}) + \lambda_{\text{TV}} R_{\text{TV}}(\mathbf{x}). \tag{9}$$

**LM.** Singh et al. (2021) investigated white-box, optimization based data reconstruction, improving the quality by applying a deep image prior (Ulyanov et al., 2018) to regularize $\mathbf{x}$. They used a CNN model as the prior, synthesizing $\mathbf{x} = f_\theta(n)$ from noise $n \sim \mathcal{N}(\mathbf{0}, \mathbf{I})$, with the noise fixed after initialization:

$$\min_{\theta} d_{\mathcal{H}}(f_c(\mathbf{x}), \mathbf{H}_{\text{private}}) + \lambda_{\text{TV}} R_{\text{TV}}(\mathbf{x}). \tag{10}$$

**GLASS.** (Li et al., 2024) considered a scenario where the adversary has knowledge of the test data distribution, and available to collect auxiliary data to train a StyleGAN. Instead of directly updating $\mathbf{x}$, the adversary updates the latent code $\mathbf{z} \in \mathcal{Z}$ or $\mathbf{w} \in \mathcal{W}^+$ to enhance the quality of the generated image. In first stage, they randomly initialize $\mathbf{z}$ then update it:

$$\min_{\mathbf{z}} d_{\mathcal{H}}(f_c(G(\mathbf{z})), \mathbf{H}_{\text{private}}) + \lambda_{\text{TV}} R_{\text{TV}}(\mathbf{x}) + \lambda_{\text{KL}} R_{\text{KL}}(\mathbf{z}). \tag{11}$$

After several updates, they obtain $\mathbf{w}+ = f_{\text{mapping}}(\mathbf{z})$ through the StyleGAN's mapping network $f_{\text{mapping}} : \mathcal{Z} \to \mathcal{W}+$, and proceed to update $\mathbf{w}+$ for fine-grained reconstruction:

$$\min_{\mathbf{w}+} d_{\mathcal{H}}(f_c(G(\mathbf{w}+)), \mathbf{H}_{\text{private}}) + \lambda_{\text{TV}} R_{\text{TV}}(\mathbf{x}). \tag{12}$$

**GradViT.** Hatamizadeh et al. (2022) studied a reconstruction problem distinct from ours. They focused on gradient inversion, aiming to reconstruct training data in the context of federated learning. They proposed a regularization method specifically designed for ViT:

$$\begin{aligned} R_{\text{patch}}(\mathbf{x}) = &\sum_{k=1}^{\frac{H}{P}-1} \|\mathbf{x}[:, P \cdot k, :, :] - \mathbf{x}[:, P \cdot k - 1, :, :]\|_2 \\ &+ \sum_{k=1}^{\frac{W}{P}-1} \|\mathbf{x}[:, :, P \cdot k, :] - \mathbf{x}[:, :, P \cdot k - 1, :]\|_2, \end{aligned} \tag{13}$$

where $P$ is the patch size of the ViT model. This regularization term aims to smooth the edges between patches, achieving higher quality reconstruction.

## B    DEFENSIVE ALGORITHMS

**NoPeek.** (Vepakomma et al., 2020) studies the model privacy leaks, and then propose to minimize privacy leaks by traning the model to minimize distance correlation between $\mathcal{X}$ and $\mathcal{H}$.

$$\min_{\theta} \mathbb{E}[\text{dCOR}(f_c(\mathbf{x}; \theta), \mathbf{x})]. \tag{14}$$

**DISCO.** (Singh et al., 2021) considers to prune out ratio $r$ of IRs' channels to reduce information leaks with an auxiliary dynamic channel pruning module $f_p$.

$$\mathbf{H}'_{\text{private}} = f_p(\mathbf{H}_{\text{private}}, r), \tag{15}$$

where $f_p$ is trained with adversarial approach, using an adversarial reconstructor $f_r$.

$$\begin{aligned} L_{\text{util}} &= \mathbb{E}[\ell_{\text{util}}(f_s(\mathbf{H}'_{\text{private}}), y)], \\ L_{\text{privacy}} &= \mathbb{E}[\|f_r(f_s(\mathbf{H}'_{\text{private}})) - \mathbf{x}\|_2], \\ &\min_{f_p}[\max_{f_r} -L_{\text{privacy}} + \rho \min_{f_c, f_s} L_{\text{util}}]. \end{aligned} \tag{16}$$

Table 3: Hyperparameters for the optimization-based reconstruction.

|                      | rMLE                          | LM                                   | GLASS                         |
|----------------------|-------------------------------|--------------------------------------|-------------------------------|
| Variable init.       | $x = \mathbf{0}$              | $x \sim \mathcal{N}(\mathbf{0}, \mathbf{I})$ | $z \sim \mathcal{N}(\mathbf{0}, \mathbf{I})$ |
| Optimizer            | Adam (lr $= 0.05$)            | Adam (lr $= 0.01$)                   | Adam (lr $= 0.01$)            |
| Total iterations ($n$) | 20,000                      | 20,000                               | 20,000                        |
| Pretrained model     | -                             | -                                    | StyleGAN2-ADA (FFHQ)          |
| $R_{\text{TV}}$      | 1.5                           | 0.05                                 | 0.03                          |
| $R_{\text{KL}}$      | -                             | -                                    | 1.0                           |
| $R_{\text{patch}}$   | 0.001                         | 0.001                                | 0                             |

|                      | DRAG                          | DRAG++                               |
|----------------------|-------------------------------|--------------------------------------|
| Variable init.       | $\mathbf{z}_T \sim \mathcal{N}(\mathbf{0}, \mathbf{I})$ | $\mathbf{z}_t = \sqrt{\alpha_t}\mathbf{z}_0 + \sqrt{1 - \alpha_t}\epsilon, \epsilon \sim \mathcal{N}(\mathbf{0}, \mathbf{I})$ |
| Sampler              | DDIM($w = 1,000$)             | DDIM($w = 1,000$)                    |
| Sampling steps ($T$) | 250                           | $250 \times 0.3 = 75$                |
| Self-recurrence ($k$)| 64                            | 64                                   |
| Noise scale ($t$)    | 1.0                           | 0.3                                  |
| Pretrained model     | SD v1.5                       | SD v1.5                              |
| $R_{\text{TV}}$      | 0                             | 0                                    |
| $R_{\text{patch}}$   | 0                             | 0                                    |
| $\lambda_1$          | 0.02                          | 0.02                                 |
| $\lambda_2$          | 0.5                           | 0.5                                  |

Table 4: Hyperparameters for the learning-based reconstruction.

|                      | Performance                                                  |
|----------------------|-------------------------------------------------------------|
| Optimizer            | Adam (lr $= 0.001$)                                          |
| Scheduler            | Cosine annealing w/ warm restart (warm-up = 5000 iterations) |
| Total iterations ($n$) | 50,000                                                    |
| Batch size           | 256                                                         |

## C  TRAINING DETAILS, EXPERIMENT ENVIRONMENT AND TIME COST

We provide the hyperparameters for various optimization-based and learning-based reconstruction attack in Table 3 and Table 4, respectively.

The experiments were conducted on a server equipped with 384 GB RAM, two Intel Xeon Gold 6226R CPUs, and eight NVIDIA RTX A6000 GPUs. To record time costs, we averaged the time cost to the experiments at the deepest split point (layer 12) over 5 samples: 2 hours 15 minutes and 41 seconds ± 15.51 seconds.

## D  ADDITIONAL EXPERIMENTS

### D.1  EVALUATION OF OPTIMIZATION-BASED ATTACKS ON THE FFHQ DATASET

We present the results conducted on the FFHQ dataset, which aligns with the assumption that GLASS has the knowledge of the private data distribution. This allows the adversary to collect an auxiliary dataset $D_aux$ from public sources and train a StyleGAN[4].

---

[4]We adapted the official StyleGAN2 ADA release from https://github.com/NVlabs/stylegan2-ada-pytorch

Table 5: Performance of the optimization-based reconstruction at different split points, evaluated using only the FFHQ dataset.

| Split Point | Method | MSE (↓) | PSNR (↑) | SSIM (↑) | LPIPS (↓) | DINO (↑) |
|---|---|---|---|---|---|---|
| Layer 0 | GLASS | **0.0017** | **28.8557** | **0.8907** | **0.0554** | **0.8751** |
| | DRAG | 0.0037 | 24.8653 | 0.8106 | 0.1181 | 0.7875 |
| | DRAG++ | 0.0036 | 24.9234 | 0.8091 | 0.1152 | 0.8073 |
| Layer 3 | GLASS | 0.0117 | **24.9676** | **0.8365** | **0.0476** | **0.9495** |
| | DRAG | 0.0057 | 22.8811 | 0.7610 | 0.0944 | 0.8820 |
| | DRAG++ | **0.0054** | 23.1041 | 0.7620 | 0.0952 | 0.8863 |
| Layer 6 | GLASS | 0.0223 | 19.7976 | 0.6731 | **0.0897** | **0.9411** |
| | DRAG | 0.0111 | 20.0196 | 0.6753 | 0.1256 | 0.8870 |
| | DRAG++ | **0.0093** | **20.8714** | **0.7620** | 0.0952 | 0.8923 |
| Layer 9 | GLASS | 0.0385 | 16.6343 | 0.5329 | 0.1733 | **0.8908** |
| | DRAG | 0.0184 | 17.9911 | 0.6005 | **0.1566** | 0.8792 |
| | DRAG++ | **0.0132** | **19.3084** | **0.6285** | 0.1823 | 0.8652 |
| Layer 12 | GLASS | 0.1153 | 10.2516 | 0.3389 | 0.3740 | 0.7099 |
| | DRAG | 0.0454 | 14.5935 | 0.4986 | 0.2464 | 0.7940 |
| | DRAG++ | **0.0191** | **17.7198** | **0.5851** | **0.2407** | **0.8207** |
| Embedding | GLASS | **0.0733** | **11.5428** | **0.2937** | **0.4139** | **0.7453** |
| | DRAG | 0.0995 | 10.2532 | 0.2834 | 0.5848 | 0.7436 |

Table 6: Performance of learning-based attacks that solely use $f_r$ to decode the IR in CLIP-ViT-B/16 at different split points, with no defenses applied.

| Dataset | Split Point | MSE (↓) | PSNR (↑) | SSIM (↑) | LPIPS (↓) | DINO (↑) |
|---|---|---|---|---|---|---|
| ImageNet → ImageNet | Layer 0 | 0.0001 | 40.1771 | 0.9880 | 0.0002 | 0.9999 |
| | Layer 3 | 0.0019 | 27.1763 | 0.8491 | 0.0826 | 0.9251 |
| | Layer 6 | 0.0071 | 21.4938 | 0.5969 | 0.3580 | 0.6740 |
| | Layer 9 | 0.0137 | 18.6311 | 0.4632 | 0.5102 | 0.4388 |
| | Layer 12 | 0.0198 | 17.0440 | 0.4044 | 0.5617 | 0.3306 |
| ImageNet → FFHQ | Layer 0 | 0.0000 | 49.7745 | 0.9975 | 0.0000 | 1.0000 |
| | Layer 3 | 0.0004 | 34.2224 | 0.9495 | 0.0236 | 0.9499 |
| | Layer 6 | 0.0028 | 25.5300 | 0.7738 | 0.1901 | 0.7901 |
| | Layer 9 | 0.0082 | 20.8773 | 0.6218 | 0.3943 | 0.5805 |
| | Layer 12 | 0.0161 | 17.9304 | 0.5263 | 0.5017 | 0.4534 |

## D.2 DATA EFFICIENCY OF THE AUXILIARY RECONSTRUCTION NETWORK

We report the performance of $f_r$ across various split points without applying any defenses in Table 6. Additionally, we report the performance when training $f_r$ on a smaller dataset in Table 7. The subset is obtained by randomly sampling from the original dataset. The number indicated below the dataset name represents the size of the subset used for training and validation (with validation employed solely for selecting the best checkpoint).

Table 7: Performance of learning-based attacks that solely use $f_r$ to decode the IR in CLIP-ViT-B/16 at different split points, training with different dataset sizes, with no defenses applied.

| Dataset | Split Point | MSE ($\downarrow$) | PSNR ($\uparrow$) | SSIM ($\uparrow$) | LPIPS ($\downarrow$) | DINO ($\uparrow$) |
|---|---|---|---|---|---|---|
| ImageNet $\rightarrow$ ImageNet (512467:128116) | Layer 6 | 0.0071 | 21.4938 | 0.5969 | 0.3580 | 0.6735 |
| | Layer 9 | 0.0137 | 18.6311 | 0.4632 | 0.5102 | 0.3503 |
| | Layer 12 | 0.0410 | 13.8742 | 0.3111 | 0.6373 | 0.2543 |
| ImageNet $\rightarrow$ ImageNet (40000:10000) | Layer 6 | 0.0095 | 20.2271 | 0.5553 | 0.3523 | 0.6187 |
| | Layer 9 | 0.0232 | 16.3395 | 0.4028 | 0.5096 | 0.3481 |
| | Layer 12 | 0.0345 | 14.6217 | 0.3431 | 0.5849 | 0.2800 |
| ImageNet $\rightarrow$ ImageNet (4000:1000) | Layer 6 | 0.0188 | 17.2608 | 0.4399 | 0.3778 | 0.3805 |
| | Layer 9 | 0.0388 | 14.1170 | 0.3032 | 0.5545 | 0.2145 |
| | Layer 12 | 0.0576 | 12.3988 | 0.2396 | 0.6385 | 0.1772 |

