# OpenReview forum: "DRAG: Data Reconstruction Attack using Guided Diffusion"
_ICLR.cc/2025/Conference — ICLR 2025 Conference Withdrawn Submission_

### Official Review · Reviewer_Q51A · 2024-11-03

**Soundness:** 3
**Presentation:** 3
**Contribution:** 3
**Rating:** 6
**Confidence:** 2

**Summary:**

In this paper, the authors aim at the data privacy in split inference, and propose a data reconstruction attack, which is based on guided diffusion model. It leverages the rich prior knowledge embedded in a later diffusion model pertained on a large-scale dataset. Experiments are conducted to verify the effectiveness.

**Strengths:**

1. The studied problem is useful because the privacy is an important in AI.
2. The proposed method seems reasonable.
3. The paper is clearly written.

**Weaknesses:**

1. Actually, there exist a lot of  image reconstruction methods, the authors  should compare with these methods. Thus, readers can better understand the advantage of the proposed method.
2. The authors should discuss whether the proposed method can be used to the CNN architecture.

**Questions:**

see the weakness

---

### Official Review · Reviewer_HJ1o · 2024-11-03

**Soundness:** 2
**Presentation:** 3
**Contribution:** 2
**Rating:** 5
**Confidence:** 3

**Summary:**

The paper presents a novel attack that exploits the prior knowledge captured by latent diffusion models (LDMs) to reconstruct input data from deep-layer intermediate representations. The findings demonstrate that this attack is capable of effectively reconstructing data from widely utilized vision foundation models, particularly CLIP, thus highlighting a significant privacy threat even in general-purpose vision encoders. Additionally, the authors explore various defense strategies specifically tailored for vision transformers, aimed at mitigating the risk of privacy leakage. This exploration of defenses is a valuable contribution to the ongoing discourse on privacy in machine learning models.

**Strengths:**

1. Compared to existing methods like GLASS, DRAG successfully reconstructs data across multiple datasets, demonstrating strong generalization performance on these datasets.
2. DRAG excels in reconstructing data from deep-layer intermediate representations, maintaining good reconstruction quality even in scenarios with limited information.
3. The designed experiments systematically evaluate the attack performance of models under privacy protection mechanisms, providing an in-depth analysis of the impact of different defense algorithms (such as DISCO and NoPeek) on reconstruction outcomes, thereby enhancing the understanding of privacy risks in practical applications.

**Weaknesses:**

1. The research primarily focuses on the vulnerabilities of the CLIP-ViT-B/16 model but does not assess the effectiveness of the attack on a broader range of architectures. The authors might consider discussing the performance of DRAG on other architecture models or models of different sizes.
2. The paper may require a more detailed discussion of the metrics used to evaluate reconstruction quality. Current evaluations seem to focus primarily on fidelity without fully considering the potential impact of data reconstruction on user privacy.
3. The attack is based on the assumption that the adversary has white-box access to the model architecture and parameters. In practical scenarios, this may not always hold true.
4. Although the paper explores various defense strategies, it lacks an in-depth analysis of their effectiveness. Additionally, the proposed method involves complex processes, such as using auxiliary datasets and multiple reconstruction steps, which may hinder its practical implementation.

**Questions:**

Please refer to Weaknesses.

---

### Official Review · Reviewer_VbeE · 2024-11-04

**Soundness:** 3
**Presentation:** 3
**Contribution:** 3
**Rating:** 5
**Confidence:** 2

**Summary:**

The paper proposes DRAG, an advanced white-box data reconstruction attack that leverages the rich prior knowledge embedded in a pretrained latent diffusion model (LDM). It can be performed in two settings. The optimization-based reconstruction setting optimizes the input latent z_t of the LDM so that the generated image can produce a feature vector similar to the target one. The learning-based reconstruction setting (DRAG++) trains an inversion network based on prior knowledge about the target network and uses the inverted result to initialize the follow-up optimization-based reconstruction. DRAG and particularly DRAG++ outperform previous baselines in case the split point is in a very deep layer.

**Strengths:**

- The algorithm is simple and reasionable.
- DRAG++ outperforms previous baselines in case the split point is in a very deep layer. It effectively bypasses different defense mechanisms.

**Weaknesses:**

- The paper did not define DRAG++. I just assume it is the learning-based + optimization-based scheme.
- I am a bit unclear about Equation 8. What is the value range of x? If x is in the [0, 1] range, $max(x^2 - 1, 0)$ should be simplified as $1 - x^2$. If x is in a larger value range, e.g., [0, 255], I do not understand the purpose of its second term.
- Why isn't DRAG++ tested when the split point is "Embedding"?
- From the qualitative results in Fig. 3, the algorithms are still hit-and-miss. In most of the tests, DRAG++ is better than DRAG, but in one setting (Fig. 3c), DRAG is better.
- DRAG seems slow. According to the Appendix, it requires more than 2 hours to process 5 samples.

**Questions:**

See weaknesses.

---

### Official Review · Reviewer_BdZf · 2024-11-04

**Soundness:** 2
**Presentation:** 1
**Contribution:** 2
**Rating:** 5
**Confidence:** 3

**Summary:**

The paper introduces DRAG, a Data Reconstruction Attack based on Guided Diffusion designed for split inference (SI) scenarios. DRAG specifically targets the reconstruction of data in vision transformers, a problem unexplored by previous methods. By leveraging the prior knowledge embedded in large latent diffusion models, DRAG effectively reconstructs input data. Experimental results demonstrate that DRAG outperforms existing methods in reconstructing data from deeper layers of vision transformers.

**Strengths:**

* The paper aims to address the data reconstruction in vision transformers.

* Experimental results demonstrate that DRAG outperforms existing methods in reconstructing data from deeper layers of vision transformers.

**Weaknesses:**

* The paper aims to address the problem of reconstructing data in vision transformers  in the context of split inference (SI). However, DRAG seems to be not specific to vision transformers. i.e. it can work with any target model architecture (see Algorithm 1). The authors should elaborate on how DRAG specifically addresses the challenges of data reconstruction in vision transformers.


* The experimental setup is unclear. The model's utility and privacy are evaluated on a validation set, $D_{val}$. However, the authors do not provide any information about  $D_{val}$. It's unclear whether $D_{val}$ is a subset of the training data or if it's derived from external datasets. A more detailed explanation of $D_{val}$’s source and relationship to the training data would clarify the evaluation process.

* The authors present results for two variants: DRAG and DRAG++. While DRAG++ is used in the tables and figures, a detailed description of DRAG++ is absent from the main text.

* Table 2 is missing some numbers of Acc.

Overall, the writing of the paper should be improved. The paper's abstract, introduction, and methodology should be consistent and coherent. Additionally, the presentation should be enhanced to address missing numbers and typos.

**Questions:**

* What is DRAG++? Why does Figure 2 not have the images of DRAG++?

* In the experiment setups,  $D_{public}$ is divided from the same dataset as $D_{priv}$. As a result, the gap between $D_{public}$ and $D_{priv}$ is close. I am curious to see how the proposed method performs in distribution shift, i.e,  $D_{public}$ is from another dataset.

* Although the proposed method aims to attack on large models, does the proposed method perform well on smaller models like ResNet?

* What is the task of the target model trained on FFHQ? Is it a classification task?

* Could the author provide the details of $D_{val}$?

* Figure 3 appears to have some typos. Should it be 'DRAG' instead of 'DRGA'?

* Please complete the missing accuracy values (Acc) in Table 2.

---

### Comment · Area_Chair_vjyG · 2024-11-22

Dear Authors and Reviewers,

The discussion phase has passed 10 days. If you want to discuss this with each other, please post your thoughts by adding official comments.

Thanks for your efforts and contributions to ICLR 2025.

Best regards,

Your Area Chair

---

### Note · Authors · 2024-11-27

**Comment:**

Thank you for the valuable feedback and insights provided during the review process. After careful consideration, we have decided to withdraw the submission at this stage to focus on further revising and improving the paper.

**Withdrawal Confirmation:**

I have read and agree with the venue's withdrawal policy on behalf of myself and my co-authors.